# LiDAR-as-Camera for End-to-End Driving

**DOI:** 10.3390/s23052845

**Published:** 2023-03-06

**Authors:** Ardi Tampuu, Romet Aidla, Jan Aare van Gent, Tambet Matiisen

**Affiliations:** Institute of Computer Science, University of Tartu, 51009 Tartu, Estonia

**Keywords:** autonomous driving, end-to-end driving, LiDAR in autonomous driving, evaluation, generalization

## Abstract

The core task of any autonomous driving system is to transform sensory inputs into driving commands. In end-to-end driving, this is achieved via a neural network, with one or multiple cameras as the most commonly used input and low-level driving commands, e.g., steering angle, as output. However, simulation studies have shown that depth-sensing can make the end-to-end driving task easier. On a real car, combining depth and visual information can be challenging due to the difficulty of obtaining good spatial and temporal alignment of the sensors. To alleviate alignment problems, Ouster LiDARs can output surround-view LiDAR images with depth, intensity, and ambient radiation channels. These measurements originate from the same sensor, rendering them perfectly aligned in time and space. The main goal of our study is to investigate how useful such images are as inputs to a self-driving neural network. We demonstrate that such LiDAR images are sufficient for the real-car road-following task. Models using these images as input perform at least as well as camera-based models in the tested conditions. Moreover, LiDAR images are less sensitive to weather conditions and lead to better generalization. In a secondary research direction, we reveal that the temporal smoothness of off-policy prediction sequences correlates with the actual on-policy driving ability equally well as the commonly used mean absolute error.

## 1. Introduction

Fully end-to-end autonomous driving systems rely on a neural network to transform sensory inputs into raw driving commands without clearly defined sub-modules [1,2,3]. This contrasts with the modular approach, which divides driving into many separately solved and evaluated tasks, such as object detection, localization, path planning, and path-tracking control [4]. End-to-mid approaches are a compromise between the end-to-end and modular approaches. The raw inputs are transformed into either a desired trajectory [5,6] or a set of occupancy or cost maps [7,8,9], based on which a safe trajectory can be computed. Essentially, in end-to-mid approaches, the neural network takes care of the perception, scene understanding, and planning, while the job of following the planned trajectory is outsourced to a control algorithm. There is a variety of path-tracking control algorithms for autonomous driving to choose from [10,11]. In this work, we follow the fully end-to-end paradigm, and our network transforms the inputs directly into the desired steering angles.

End-to-end and end-to-mid autonomous driving is often achieved based solely on camera images [5,12,13,14] or with additional information about the desired route provided via one-hot encoded navigation commands or via route planner screen images [15,16,17]. This input combination is cheap and seems sufficient for human drivers to complete routes safely, making it an interesting subject of study. However, in simulation, a precise depth image can readily be generated and has been shown to be useful for driving models [18]. Even an approximate depth image predicted from an RGB image may improve results [18,19].

In the real world, one can also attempt to predict depth images based on monocular camera images [20,21,22]. Alternatively, stereo cameras can be used, but they suffer from a limited range. A more reliable depth image can be obtained by projecting the LiDAR point cloud to an RGB camera image. However, merging this depth image with a camera image is not trivial [23]. The two sensors are not located in the same place and may see the world from different angles, leading to different blind spots. Furthermore, it is difficult to guarantee temporal synchronization of the camera and the LiDAR. Except for the blind spots, extrinsic calibration of the sensors allows a decent matching of the depth and color values (either RGB-D images or colors attached to point cloud points). However, calibration is a time-consuming procedure that may need to be repeated regularly. Finally, even if good calibration is achieved for the training data, unexpected synchronization failures or calibration drifts may occur during deployment [24].

To remove the need to align multiple sensors, Ouster LiDARs allow the generation of a surround-view image containing perfectly aligned depth, intensity, and ambient radiation channels [25]. The intensity and ambient radiation channels can be seen as providing visual information but from the infrared wavelength range. This three-channel input, therefore, contains temporally and spatially perfectly aligned visual (infrared) and depth information.

This input is in image form and can be analyzed using any of the very successful approaches developed in computer vision and camera-based self-driving. Network architectures for extracting information from images are more mature than those for point clouds. We hence have a sensor providing seemingly rich information in a form that we know well how to analyze. What intuitively seems informative, however, may be too unreliable for basing a self-driving solution on it. The main goal of the present work is to evaluate the usefulness of these LiDAR images as model inputs for end-to-end self-driving.

Our main contributions are the following:We demonstrate that LiDAR images, as produced by Ouster OS1-128 LiDAR firmware, contain sufficient information for road-following on complex and narrow rural roads, hence validating their usefulness for self-driving.We compare LiDAR-image-based driving with camera-based driving and show it adds robustness to light and weather conditions in this task.We study the correlation between off-policy and on-policy performance metrics, which has not been studied before in a real car context.We collect and publish a real-world dataset of more than 500 km of driving data on challenging rural roads, with LiDAR and camera sensors and centimeter-level accurate GNSS trajectory. The dataset covers a diverse set of weather conditions, including snowy winter conditions.

This is the first experiment in our effort to validate the usefulness of these LiDAR images for increasingly complicated driving tasks, such as highway driving and urban driving. Here, we restrict ourselves to the simpler task of road following, albeit in the complex setting of narrow rural roads, which are also used as World Rally Championship tracks, meaning they are also challenging for humans.

## 2. Methods

In this section, we will introduce the experimental design applied to evaluate the usefulness of the novel LiDAR image inputs for end-to-end self-driving. In particular, after giving a definition of the behavioral cloning paradigm commonly used in end-to-end driving, we describe the dataset and the neural network models used. The key idea is to train LiDAR image-based models exactly the same way as if the inputs were visible light RGB images and to compare how such models perform when deployed. We introduce the deployment procedure and the performance metrics we use.

### 2.1. Behavioral Cloning

Behavioral cloning takes a supervised learning approach to self-driving [26]. Based on information from a chosen set of sensors, the model is optimized to produce the same driving behavior as a human would. This behavior is usually described by the sequence of low-level commands given, or the trajectory taken [1,2,3]. For model training, a dataset is collected consisting of sensor recordings during human driving, accompanied by the driving commands, or the trajectory produced by the driver.

Such an imitation approach has worked well for simpler tasks such as lane following [12,26], which seem to not require restrictive amounts of training data. However, dense traffic scenarios remain challenging for behavioral cloning [27]. In addition to Tesla and comma.ai, multiple companies report promising performance in real-world urban driving with neural network-based solutions [17,28,29], but it is unclear to what extent these can be considered end-to-end. Though end-to-end models in other fields, e.g., speech recognition, have shown good generalization, replicating this success in self-driving is costly due to the massive amounts of data the cars produce. As further limitations, safety guarantees against rare situations and adversarial attacks are lacking [30,31], and interpreting model decisions remains challenging [1,32].

### 2.2. Data Collection

In the period of May 2021 to October 2021, training recordings of human driving were collected from all non-urban WRC Rally Estonia tracks and a few similar routes. Driving was performed with Lexus RX 450h fitted with a PACMod v3 drive-by-wire system provided by AutonomouStuff. The following sensors were recorded: NovAtel PwrPak7D-E2 GNSS device, Ouster OS1-128 LiDAR, three Sekonix SF3324 120-degree FOV cameras, and one Sekonix SF3325 60-degree FOV camera (Figure 1). All tracks were recorded in both directions at least once, amounting to more than 500 km of driving. The road type was mostly very low-traffic gravel roads. There were shorter sections of two-lane paved roads. In January–February 2022 and in May 2022, further data collection was performed in snowy and early spring conditions. These data were only used for off-policy metric computation, not for training. The list of recordings used in this work is detailed in Appendix D.

The driving recordings from spring, summer, autumn, and winter differ strongly in vegetation levels and light conditions. All driving was completed in daylight but in differing weather conditions, including heavy rain. The dataset, including recordings from sensors not used in this work, will be made fully available with this publication.

### 2.3. Data Preparation

For neural network training, only recordings from the Ouster OS1-128 mid-range LiDAR and Sekonix SF3324 RGB camera placed front center of the car were used. The list of recordings from May to October 2021 was divided into training (460 km of driving) and validation sets (80 km of driving). Recordings from the evaluation track, where the on-policy evaluation was later performed, were not part of the training set unless stated otherwise but were part of the validation set that was used for early stopping. The lists of recordings used for model training and validations are given in Appendix D.

The surround-view LiDAR image output of Ouster OS1-128 LiDAR contains the channels of depth, intensity, and ambient radiation (Figure 2). In this work, the depth channel is further pre-processed in a way that distances in the range from 0 to 50 m are mapped linearly to the values 255 to 0, i.e., 20 cm depth resolution. All distances beyond 50 m are marked as 0. For both RGB and LiDAR image inputs, the image was cropped horizontally to remove the hood of the car and all rows above the horizon. For both input types, the image was cropped vertically to keep 90 degrees of view in the center front. The camera image was also downscaled to make it match the LiDAR image size. No further processing was performed. This resulted in a 264×68×3 image as neural network input for both input types. The target labels correspond to the steering wheel angles as produced by the human driver collecting the data.

No data diversification methods were employed because, firstly, we assume our data set is large enough to learn the task without augmentations, and secondly, useful augmentation is not easy to perform—recent works have described that models often learn to detect the fact of augmentation itself instead of learning a generalized policy [33,34].

The dataset was not balanced in any way. The rally tracks are curvy and are not dominated by stereotypical drive-straight behavior. We do not think there is any type of situation in our data that should be undersampled. Crossroads and interactions with other cars on the road are not excluded from the dataset, despite using the data only for learning road-following.

### 2.4. Architecture and Training Details

Our intent is to use a relatively simple network architecture because the goal is to compare two input modalities rather than to achieve the best possible model. With limited data amount or variability, powerful models could overfit, masking the effect of chosen inputs. Recent works have reported success on as low as 30 h of training data [17,27]. Our 500 km of data corresponds to only 15 h of driving, placing us in danger of overfitting our models.

We use a slightly modified version of the classical PilotNet architecture from [12]. We add batch normalization after each layer except the last two fully-connected layers. This gives us more training stability and faster convergence, similar to what is discussed in [35]. For similar reasons, we use LeakyReLU [36] as the activation function instead of ReLU. The architecture is summarized in Table 1. The model outputs the steering angle as the lateral control command. The driving speed is not controlled by the network.

**Table 1 sensors-23-02845-t001:** Details of the architecture. Batch normalization always precedes activation function. All convolutions are applied with no padding. The resulting output dimensions can be seen in Figure 3.

Layer	Hyperparameters	BatchNorm	Activation
Input	(264, 68, 3)		
Conv2d	filters = 24, size = 5, stride = 2	BatchNorm2d	LeakyReLU
	filters = 24, size = 5, stride = 2	BatchNorm2d	LeakyReLU
	filters = 36, size = 5, stride = 2	BatchNorm2d	LeakyReLU
	filters = 48, size = 3, stride = 1	BatchNorm2d	LeakyReLU
	filters = 64, size = 3, stride = 1	BatchNorm2d	LeakyReLU
Flatten	-	-	-
Linear	nodes = 100	BatchNorm1d	LeakyReLU
	nodes = 50	BatchNorm1d	LeakyReLU
	nodes = 10	none	LeakyReLU
	nodes = 1	none	none

We used mean absolute error (MAE) as the loss function, as this metric has been shown to correlate better than mean squared error with on-policy driving ability [37]. We used an Adam optimizer with weight decay [38] with default parameters in PyTorch [39]. We used early stopping if no improvement in the validation set was achieved in 10 consecutive epochs, with the maximum epoch count fixed to 100. The code with model definitions and training procedure will be made available on GitHub with the publication.

It has been reported that multiple training runs can result in clearly differing on-policy behaviors [27]. To minimize the potential effect of training instability, we train three versions of our main models and report the metrics for each.

### 2.5. Evaluation Metrics

The models were evaluated on-policy and off-policy. It is widely reported that off-policy metrics correlate poorly with actual driving ability [37]. However, they are cheap to compute before deploying the solution. If a better off-policy metric could be found, development could be accelerated by selecting only the best models for real-life evaluation.

Off-policy metrics are computed using human-driven validation recordings originating from the same season as when the on-policy evaluation happened. We limit the set of recordings to the same season because we assume that off-policy metrics computed on summer data would have no information about driving ability in the winter and vice versa.

We report the mean absolute error (MAE) between human commands and model predictions as this metric has been reported as having a favorable correlation with the driving ability [37]. In addition, following [40,41], we also compute the whiteness of the predicted command sequence:(1)$$W=\sqrt{\frac{1}{D}\sum_{i=1}^{D}{\left(\frac{\delta P_{i}}{\delta t}\right)}^{2}},$$
where $\delta P_{i}$ is the change in predicted steering angle, *D* is the size of the dataset, and δt is the temporal difference between decisions. δt = 0.1 for LiDAR and δt = 0.033 for camera.

Whiteness measures the mean smoothness of the sequence of commands generated and can be computed on-policy and off-policy. Here, $W_{off-policy}$ refers to the whiteness of the sequence of commands generated by a model on human-driving recordings from the evaluation track in the same season as when the on-policy testing took place. In contrast, $W_{on-policy}$ refers to the whiteness of the commands generated during model-controlled driving during evaluation.

We consider the smoothness of commands a promising metric because during on-policy testing, we observed that jerkiness of driving, i.e., temporally uncorrelated commands, seems to predict an imminent intervention. The model not responding to very similar consecutive frames in a consistent manner might reveal its inability to deal with the situation.

The number of interventions during a test route was counted, and distance per intervention (DpI) was computed as the main quality metric. The models were trained to perform route following and not to handle intersections. All interventions at intersections were removed from the count. For safety reasons, in the case of an oncoming car, the safety driver always took over the driving. Interventions due to traffic were also excluded from the intervention count.

As additional on-policy metrics, we measure the deviation of the model driving compared to a human trajectory on the same route. Locations were measured using the NovAtel PwrPak7 GNSS receiver combining the inertial navigation system (INS) data with real-time kinematic positioning (RTK), achieving centimeter-level precision. For each position in the model-driven trajectory, the offset is defined as the average distance to the two closest human trajectory points. The mean of this lateral difference along the route is reported as $MAE_{trajectory}$. We define failure rate as the proportion of time this lateral difference was above 1 meter.

### 2.6. On-Policy Evaluation Procedure

The on-policy evaluation was performed on a 4.3 km section of SS20/23 Elva track in both driving directions (cf. Results, Figure 4 for a map of the route). The speed along the route was set to 80% of the speed a human used in the same location on the route, as extracted from a prior recording of human driving. Driving at 100% human speed was attempted but was too dangerous to use with weaker models. For covering different weather conditions, it was intended to be completed in two parts: autumn and winter, but due to technical reasons, a third session in spring was needed.
In the last week of November 2021: the weather conditions and vegetation levels were very similar to the most recent training data recorded at the end of October. Due to a missing parameter in the inference code, the RGB models were run on BGR input, and the results had to be discarded. Hence, only LiDAR-based models were adequately tested in this session. Night driving was performed with dipped-beam headlights on. Results from these tests are marked with (Nov) in Table 2.In the first week of February 2022: with snow coverage on the road. This constitutes a clearly out-of-distribution scenery for the camera models. Moreover, also for LiDAR models, the surface shapes and reflectivity of snow piles differ from vegetation and constitute out-of-distribution conditions. LiDAR and camera images from summer, autumn, and winter are given in Appendix A. From this trial, marked with (Jan), we report only the driving performance with LiDAR, as the camera still operated in the BGR mode.In the first week of May 2022: early spring, which constitutes a close-to-training-distribution condition. Camera models were evaluated with the correct inference code. The location of LiDAR on the car had changed before this trial compared to the training data. LiDAR-based models underperformed during this test, despite our efforts to adjust the inputs.

In short, camera models were tested with adequate inputs only in spring 2022. LiDAR models were tested in autumn 2021 and winter 2022. The weather conditions were different during these tests, so a direct comparison of values should not be made.

The rally tracks are narrow and bordered by objects harmful to the car; hence, the safety driver was at liberty to take over whenever they perceived danger. An intervention is hence defined as a situation where the safety driver perceived an excessive threat to the car or the passengers. An intervention was triggered by the safety driver applying force to turn the steering wheel. If the model turned the steering wheel at the same moment and in the same direction as the safety driver, no force was applied, and no intervention was counted.

## 3. Results

In this section, we present the driving ability of our models as measured by on-policy metrics. We also compute off-policy metrics but only with the purpose of evaluating their correlations with on-policy performance. Observations on model sensitivity to inputs are given in Appendix F.

### 3.1. Driving on an Unseen Track

The ultimate goal of end-to-end self-driving is to create models that can generalize to new roads without the use of high-definition maps. Hence, we first summarize the models’ ability to generalize to the evaluation track from other similar roads. Three instances of LiDAR models were tested in autumn, and three camera models were tested in spring. Using multiple models allows the reader to grasp the stability of the results. A larger number of repetitions was not used due to the complexity of real-world evaluation. The metrics for these six evaluations are given in the first section of Table 2. The interventions during these six trial runs are also visualized on the map of the route in Figure 4.

The results indicate that in in-distribution weather, but on a novel route, the performance of LiDAR-based models is similar to or better than camera models. The evaluations took place half a year apart, but conditions were suitable in both cases. Spring testing was performed on a largely cloudy day, with only short periods of direct sunlight. The autumn test took place in cloudy and dim daylight with short periods of very light rain. These conditions should be sufficiently close to ideal for camera and LiDAR models, respectively.

### 3.2. Overfitting Setting

We next asked to what extent the task was more difficult due to needing to generalize to a new route. We trained camera and LiDAR models that included, in their training set, one human driving recording in each direction from the evaluation track. As these models will be exposed to the objects and types of turns on the evaluation track, we call this the “overfitting” (to the evaluation track) setting. The second section of Table 2 indicates that while the LiDAR model clearly benefited from test-track recordings, the effect is weaker for the camera-based model. The overfitted LiDAR model drove without interventions, while the RGB model yielded similar performance to the non-overfit models.

We conclude that approximately 500 km of road-following data in the original training set did not suffice for good generalization to similar but unseen roads. Data augmentation techniques could be applied or more data collected to increase generalization over this source of data variability.

### 3.3. Night Driving and Winter Driving

The third set of on-policy tests evaluates the models’ ability to generalize to weather conditions very different from the training distribution. We have no a priori expectation of camera-based driving models generalizing to these conditions. The camera images during the night differ drastically from daylight driving, despite using headlights. Similarly, the color distribution and brightness of camera images in the winter with snow coverage are clearly out-of-distribution. These differences are easy to detect with the human eye.

In contrast, the extent that these two novel conditions are out-of-distribution for LiDAR-based models is difficult to estimate with the naked eye. A priori, we can assume that depth and intensity channels should be affected only minimally by the lack of sunlight in night driving, with ambient radiation somewhat affected. Snow coverage adds more smooth surfaces to the landscape, but the resulting depth image may remain in proximity to the diversity of scenes contained in the training data. Ambient radiation and intensity images are likely out-of-distribution due to the different reflective properties of snow and vegetation, but the extent of its effect on LiDAR-based driving models was unknown before being tested.

The results from these trials are marked with night and winter in Table 2. Remarkably, LiDAR models trained with day-time data sets see only a minimal drop in performance when deployed at night. The camera models, however, immediately steered the car off the road, so there is no performance to report. While these experiments were performed with the flawed BGR input to the RGB models, judging from the performances in other experiments, we do not expect the performance with correct input to be much different (cf. all experiments in Appendix E).

When deploying models trained on data from spring, summer, and autumn to snow-covered roads, LiDAR-based models also see a clear drop in performance. LiDAR models manage to maintain some of their driving ability but drop from ≈4000 to 226–425 m per intervention. Qualitatively, we report that LiDAR models drove reasonably well in the forest where depth information was abundant but failed to stay on the road on sections between open fields (cf. Appendix C). As expected, RGB-based models failed to generalize to snowy roads and steered off the road immediately.

### 3.4. Informativeness of Individual LiDAR Channels

We also performed on-policy testing of models trained on individual LiDAR image channels. This was conducted to obtain a better understanding of the usefulness of each of these channels. This evaluation was performed in in-distribution weather in November 2021. As these experiments were performed on another day compared to the tables above, a three-channel model was also re-evaluated to confirm the conditions were similar. The tested models were trained with no recordings from the evaluation track in the training set.

The examples of the images from these three channels in summer, autumn, and winter are given in Appendix B. At visual inspection, the intensity channel seems approximately as sharp and as informative as a gray-scale camera image, albeit capturing a different wavelength. The depth image is less spatially dense but clearly informative about sufficiently large obstacles. However, ambient radiation images depend strongly on sunlight being present and seem an unreliable source of information.

In Table 2, we observe that the model trained based on the intensity channel can perform surprisingly well. However, neither depth nor ambient radiation channels contain sufficient information for safe driving. The depth-based model also struggled to drive safely in the forest, where trees could have provided depth cues of where to steer towards. These channels may nevertheless still contribute useful information to the three-channel model.

### 3.5. Correlation Study between On- and Off-Policy Metrics

In this work, we trained a total of 11 models (3+1 LiDAR, 3+1 camera, and individual LiDAR channels). We deployed these models in more and less suitable conditions, including accidentally deploying RGB-image models using a BGR video stream and deploying LiDAR models after the sensor location had been changed. Here, we wished to study if the on-policy performance of these test drives could have been predicted before deployment via off-policy metrics or at least during the drive via non-discrete on-policy metrics.

In the following, we used metrics from 17 model deployments. The list of trials included and the associated metrics are given in Appendix E. This list includes using RGB-based models with a BGR camera stream and using LiDAR in a changed location because the models were capable of driving despite the disadvantageous conditions. Each deployment was matched with an off-policy evaluation using recordings from the same track, similar seasons, and similar sensor configuration (e.g., using BGR images). We computed the Pearson correlation [42] between DpI and the various other on- and off-policy metrics. The DpI for trials with no interventions was set to 10 km for the computations. The resulting correlation values are given in Table 3.

Matching the perception of passengers, the whiteness of effective wheel angles during the drive shows a correlation with DpI (r=−0.67). The whiteness of the model outputs $W_{on-policy}$ and the mean distance from a human trajectory show somewhat weaker correlations with DpI. The measures used here are averages over multiple kilometers, but actual danger prediction should happen on a more precise scale. Evaluating whiteness as an online predictor of end-to-end model reliability is outside the scope of this work.

Among the off-policy metrics, $W_{off-policy}$ correlates to a similar degree with intervention frequency as the MAE of steering angles. The difference between the two Pearson correlation coefficients is not significant as per a permutation test. Notice that these two metrics are very different in nature—one measuring the quality and the other temporal stability of predictions. When combining these two metrics via summation after standardization, an even higher correlation with DpI can be obtained (r=−0.82). The improvement over MAE-only correlation is, however, not statistically significant (permutation test, mean effect size −0.05, $p_{val}$ = 0.16).

To our knowledge, mean absolute error is a very commonly used off-policy metric for estimating model quality before deployment and for early stopping during model training. MAE has been shown to correlate with driving ability better than multiple other metrics [37]. Our result suggests that the whiteness of the command sequence generated on an appropriate validation set might serve as a complementary model-selection metric (cf. Section 4).

## 4. Discussion

In the present work, we collected a high-quality dataset for the end-to-end road following task in challenging rural roads used for World Rally Championship. This dataset contains driving in narrow and complicated routes during the four seasons of the temperate climate. The measurements of all sensors, including those not used here, across more than 500 km of driving are made publicly available. The driving task contained in these data differs clearly from the usual lane-following tasks on larger roads and is complex in its own way—due to narrow roads, small radius blind turns, and uneven road surface. For example, the comma.ai driver assistance system, which excels in lane keeping on inter-city roads, fails completely to drive on our evaluation route.

On two separate sensory inputs of this dataset, LiDAR image and frontal camera, we trained models to control the steering of the car. The models were evaluated off-policy and on-policy with speed fixed to 80% of human speed. We show that LiDAR image input, as produced by Ouster OS1-128 LiDAR firmware, contains sufficient information for road-following also in the complex and narrow rally tracks designed to be challenging for humans. The task is not trivial, as evidenced by the similarly-trained RGB-image-based models achieving similar performance. Curiously, models using only the intensity channel of the LiDAR image, information often discarded in point cloud analysis, also performed competitively. If this information is sufficient or useful in more complex driving tasks such as highway driving, it may merit further study.

The benefits of LiDAR-based driving become apparent when needing to generalize to new conditions. LiDAR images are more similar across weather conditions (cf. Appendix A). We hypothesize that this allows the entirety of the training data to be useful for driving in all conditions, including those not in the training data. Driving demonstrations from sunny summer days benefit LiDAR-based driving on a dark autumn night, as evidenced by our LiDAR models being able to drive in the night and, to some extent, even in the winter. In contrast, RGB-based models can not generalize to night driving. It seems that for a simple RGB-based behavioral cloning approach, demonstrations of various traffic situations need to exist in a variety of visually different conditions, increasing the data need. A higher data efficiency of LiDAR-based models would be an interesting property, at least for research institutions that cannot boast fleets of cars collecting massive amounts of data daily.

During night driving, LiDAR models can rely on intensity and depth channels, which are active sensing and independent of external light sources. The depth channel is also independent of the reflectivity of the surfaces and yields in-distribution values also with snow coverage. While depth alone was proven insufficient for safe end-to-end driving even in training conditions, it may still contribute reliable information to the three-channel models. Furthermore, we assume that the importance of depth information becomes more apparent in highway and urban driving tasks, where distance with other traffic participants must be maintained.

LiDAR information is often used in its point cloud representation. Here, using image representation allowed us to perform a fair comparison of LiDAR and RGB camera input modalities, as identical methods could be applied. We believe processing LiDAR data in image form can be useful in general because computer vision is one of the most studied topics in deep learning, and many established architectures exist for image processing. Certain architectures are empirically validated to perform various tasks in a reliable manner, and methods exist for sensitivity analysis.

Evaluating autonomous driving systems is complicated because the ability to drive safely can only be measured by deploying the model. When exploring architectures to use, data sampling techniques, or other aspects of the training procedure, one would need to deploy the models to know which techniques work best. This is extremely costly in the real world. If a combination of off-policy metrics could be found that correlates reliably with actual driving ability when deployed, only the more promising models could be selected for testing and evident failures discarded. Here we showed that among sufficiently capable models, the whiteness, i.e., smoothness of generated commands on an appropriate validation set, predicted driving ability equally well as the magnitude of errors. We hypothesize that non-smoothness reveals the models’ uncertainty about the situation—the model reacts differently to very similar inputs. In future work, we propose to evaluate the correlations of other measures of epistemic uncertainty with on-policy performance. Using a more general uncertainty measure carries the benefit of being applicable to a wider range of output modalities, e.g., trajectories and cost maps. However, these metrics capture only variance and not bias, and a trivial constant model would show perfect stability. Hence, such stability measures should be used in combination with other metrics (e.g., MAE).

## 5. Conclusions

We conclude that the LiDAR images produced by Ouster LiDARs contain sufficient information for completing at least the simplest sub-tasks of self-driving. Moreover, this input showed high stability across environmental conditions, which allowed the driving models to perform during the night and in the winter without any training data collection conducted in those conditions. This input stability, in addition to the useful depth information, is a clear advantage over visible-light cameras. We do not envision LiDAR images replacing RGB images as the dominant input to self-driving solutions, but this input type may prove useful as a fallback system or can be used in combination with RGB images. The combined usage is facilitated by both inputs being in the image form, potentially allowing early fusion.

Additionally, we conclude that we are still in search of off-policy metrics that would reliably predict the performance when the solution will be deployed. We reveal that the stability of commands over time is almost as good a predictor of deployment success as the mean magnitude of prediction errors. Hence, when picking models to deploy, one should pay attention to multiple quality metrics or seek to develop a combined metric.

## Figures and Tables

**Figure 1 sensors-23-02845-f001:**
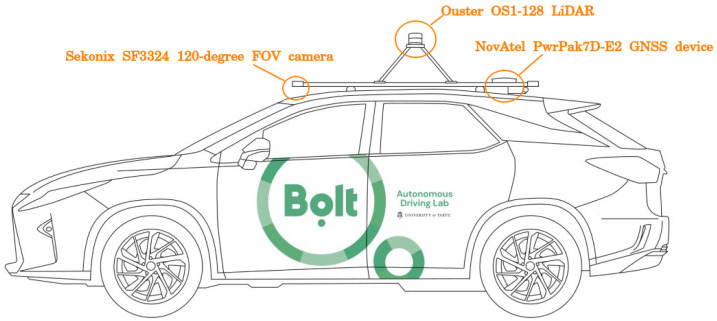
The location of sensors used in this work. There are other sensors on the vehicle not illustrated here.

**Figure 2 sensors-23-02845-f002:**
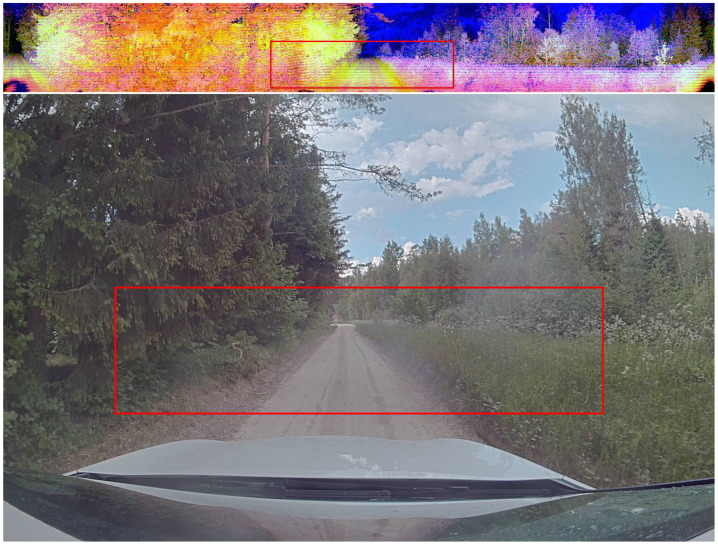
Input modalities. The red box marks the area used as model input. Top: surround view LiDAR image, with red: intensity, blue: depth, and green: ambient. Bottom: 120-degree FOV camera.

**Figure 3 sensors-23-02845-f003:**
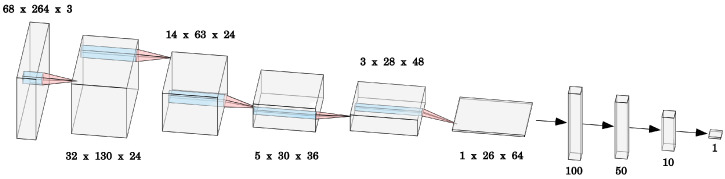
The modified PilotNet architecture. Each box represents the output from a layer, with the first box corresponding to the input of size (264, 68, 3). The model consists of 5 convolutional layers and 4 fully connected layers. The flattening operation is not made visible here. See the filter sizes, usage of batch normalization, and activation functions in Table 1.

**Figure 4 sensors-23-02845-f004:**
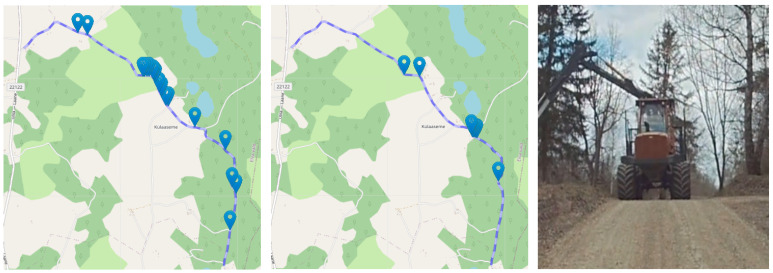
Safety-driver interventions in the experiments where the test track was not included in the training set. Interventions from 3 test runs with different versions of the same model and from both driving directions are overlaid on one map. Interventions due to traffic are not filtered out from these maps, unlike in Table 2. Left: camera models v1–v3 (first 3 rows of Table 2). Middle: LiDAR models v1–v3 (rows 4–6 of Table 2). Right: an example of a situation where the safety driver has to take over due to traffic. Such situations are not counted as interventions in Table 2.

**Table 2 sensors-23-02845-t002:** Results of on-policy evaluations. Evaluations interrupted due to a high frequency of interventions are marked with *. Horizontal lines separate values illustrating the different results subsections.

Experiment	Model (Session)	Distance	Interventions	DpI	${\mathrm{MAE}}_{\mathrm{trajectory}}$	Failure Rate	$W_{\mathrm{on}-\mathrm{policy}}$
Generalization	Camera v1 (May)	8464.33 m	2	4232.17 m	0.2309 m	0.96%	24.63 ${}^{\circ }$/s
	Camera v2 ( may)	8363.17 m	4	2090.79 m	0.2382 m	0.69%	33.46 ${}^{\circ }$/s
	Camera v3 ( may)	8389.88 m	7	1198.55 m	0.2403 m	0.66%	29.70 ${}^{\circ }$/s
	LiDAR v1 (Nov)	8442.5 m	2	4221.3 m	0.22 m	0.42%	23.0 ${}^{\circ }$/s
	LiDAR v2 (Nov)	8465.9 m	2	4233.0 m	0.24 m	0.98%	17.7 ${}^{\circ }$/s
	LiDAR v3 (Nov)	8432.3 m	3	2810.8 m	0.25 m	2.18%	18.8 ${}^{\circ }$/s
Overfitting	Camera overfit (May)	8489.14 m	3	2829.71 m	0.2453 m	1.39%	23.07 ${}^{\circ }$/s
	LiDAR overfit (Nov)	8436.9 m	0	>8436.9 m	0.26 m	4.38%	19.2 ${}^{\circ }$/s
Night	LiDAR v2 (Nov)	8216.3 m	8	1027.0 m	0.24 m	1.27%	25.6 ${}^{\circ }$/s
	LiDAR v2 #2 (Nov)	8376.6 m	3	2792.2 m	0.23 m	1.55%	20.3 ${}^{\circ }$/s
	LiDAR overfit (Nov)	8521.5 m	1	8521.5 m	0.25 m	1.52%	21.5 ${}^{\circ }$/s
Winter	LiDAR v1 (Jan)	8080.5 m	19	425.3 m	0.24 m	0.94%	38.4 ${}^{\circ }$/s
	LiDAR v2 (Jan)	8001.4 m	22	363.7 m	0.28 m	3.10%	38.7 ${}^{\circ }$/s
	LiDAR v3 (Jan)	7698.9 m	34	226.4 m	0.26 m	1.64%	42.2 ${}^{\circ }$/s
LiDAR	LiDAR v2 (Nov)	8491.6 m	0	>8491.6 m	0.22 m	1.97%	19.2 ${}^{\circ }$/s
channels	LiDAR intensity (Nov)	8446.2 m	2	4223.1 m	0.33 m	7.02%	24.0 ${}^{\circ }$/s
(next day)	LiDAR depth (Nov)	1679.0 m *	22*	76.3 m *	0.61 m *	19.95% *	29.9 ${}^{\circ }$/s *
	LiDAR ambience (Nov)	329.5 m *	19 *	17.3 m *	0.73 m *	17.49% *	168.2 ${}^{\circ }$/s *

**Table 3 sensors-23-02845-t003:** Pearson correlations of the main driving quality metric distance per intervention (DpI) with other on- and off-policy metrics. The highest-correlating metrics of both types are highlighted in bold.

	On-Policy Measures				Off-Policy Measures	
Measure	${\mathrm{MAE}}_{\mathrm{Trajectory}}$	Failure Rate	$W_{\mathrm{on}-\mathrm{Policy}}$	$W_{\mathrm{effective}}$	$W_{\mathrm{off}-\mathrm{Policy}}$	${\mathrm{MAE}}_{\mathrm{steer}}$
Pearson R	−0.56	−0.06	−0.56	**−0.67**	−0.72	**−0.76**

## Data Availability

Code and instructions for obtaining access to the data set: https://github.com/UT-ADL/e2e-rally-estonia, accessed on 20 January 2023. Othertraffic participants are not censored in the data; people and cars, including number plates, are visible. As a consequence, the data cannot be made freely available but will be made accessible upon filling out a form and agreeing to the terms of use.

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
