# Peer review of "LiDAR-as-Camera for End-to-End Driving"

_sensors, 2023, doi:10.3390/s23052845_

Round 1

Reviewer 1 Report

1- The novelty of the paper is not clear.

2- The proposed model used is not well presented.

3- A comparison with some existence proposed model should be included.

4- The paper must contain conclusion beside the discussion section.

5- Some typos have been found.

6- The paper presentation should be modified. 

Author Response

We thank the reviewer for the care and time taken to review our work. The points raised are discussed one by one below.

1- The novelty of the paper is not clear.

We thank the reviewer for raising this issue. In the original manuscript, we stated the aim to ”validate the usefulness of these LiDAR images for increasingly complicated driving tasks”. We agree this is a vague overall goal and not specific enough for the manuscript at hand.

We have now better specified the goal of the present work in the introduction and separated it from the overall goals of the research direction (lines 61-66). Notice that the novelty is also listed in the Introduction in the list of “main contributions”.

We thank the reviewer for raising this point and believe the changes have made the manuscript more to the point.

2- The proposed model used is not well presented.

We thank the reviewer for the effort put into understanding our manuscript and the work done. We find it common to present the neural network architectures in the format of a table listing the layers the model is composed of. However, we agree that while allowing replication, the table format does not give an easy intuitive understanding of the architecture. We have now added a graphical illustration of the model that complements the information in the table.

3- A comparison with some existing proposed model should be included.

We agree with the reviewer that such a comparison would be beneficial, but for the domain-specific reasons listed below, we are unable to provide it.

First of all, we would like to emphasize that the goal of the manuscript is not to propose state-of-the-art architectures or achieve state-of-the-art driving ability but to investigate if LiDAR images are sufficient to build a self-driving solution upon them and how they compare to visible light images as inputs to driving networks. Even if comparing our solutions with existing models was possible, it would not add any insight to the comparison of the usefulness of RGB vs LiDAR images.

Theoretically, there are two ways a comparison with existing solutions could be performed:

  1. Comparing on-policy metrics reported in a similar task: however, the task of road-following on narrow, one-lane rural roads is not commonly studied. This task is not comparable with lane-keeping on two-lane roads (and even less with lane-keeping on highways). We find no prior publications on sufficiently similar tasks to compare results at the level of distance per intervention.
  2. Deploying an existing model on our vehicle and evaluating it in our task: evaluating an existing model on a custom dataset is feasible in many machine learning studies but not in self-driving because most solutions are extremely sensitive to sensor placement. Applying a model to a different sensor set-up will simply not perform well. We have discussed this sensitivity as a limitation in the supplementary materials. For this reason and for the sake of protecting business secrets, real-world driving models are not commonly made available, unlike models trained in the CARLA simulator. The only exception is the commercially available comma.ai device, which we have installed on our car and tested on this task. Comma.ai fails the task completely, failing to take most of the turns. By the on-screen visualizations, it seems the comma.ai system fails to detect lane (road) boundaries in the narrow and sharp turns that are common in our task. The failure of comma.ai further emphasizes point 1); that the task is very different from the usual lane-keeping on 2-lane roads and highways.

Beyond reporting that comma.ai is unable to perform well in the on-policy evaluation, we, unfortunately, see no further comparison with existing solutions we could provide. In the manuscript, we do not wish to emphasize the failure of comma.ai as the device is clearly not designed to perform well in our task. We nevertheless now added this piece of information to the conclusions section to emphasize the task being complex and unusual.

 4- The paper must contain a conclusion beside the discussion section.

We thank the reviewer for suggesting this and agree that a short take-home message would be useful. Other reviewers also suggested something similar. We have now added this section (line 428-443).

5- Some typos have been found.

We thank the reviewer for their attention. We have now performed extensive spell and sentence structure checking. 

6- The paper presentation should be modified. 

The paper has now been modified, taking into account the above-mentioned comments by the reviewer as well as comments by other reviewers.

Reviewer 2 Report

This paper needs revision.

1) The introduction is too short. In particular, the concept, application, and development of LiDAR should be introduced, where the paper "Large-Scale Bandwidth and Power Optimization for Multi-Modal Edge Intelligence Autonomous Driving" should be mentioned.

2) Please highlight your problem, which is a key to research.

3) There are many typos.

4) Could you draw some insights for practice?

Author Response

We thank the reviewer for the time and effort put into reviewing our work. We have now modified our manuscript according to some of the suggestions. The responses to the points raised are listed below.

1) The introduction is too short. In particular, the concept, application, and development of LiDAR should be introduced, where the paper "Large-Scale Bandwidth and Power Optimization for Multi-Modal Edge Intelligence Autonomous Driving" should be mentioned.

We thank the reviewer for pointing out this interesting manuscript and for suggesting improvements to our manuscript. However, at this point, the authors maintain that LiDAR is a very common sensor in the field of autonomous driving and needs no further explanation for the experts in the field. This is supported by other articles in the field not discussing the LiDAR technology in detail. We find that for the intended audience, a paragraph about LiDAR technology will add no extra value.

2) Please highlight your problem, which is a key to research.

We thank the reviewer for raising this point. In combination with the comments from other reviewers, this has made us realize we need to specify the goal of the study in more clarity. In the original version of the manuscript, the longer-term goals of the lab were mentioned without clearly wording the object of study for this particular manuscript. 

We have now modified the introduction and improved the phrasing, making the research question clearer (lines 61-66).

3) There are many typos.

We thank the reviewer for attentive reading and for pointing out this problem. We have now performed extensive spell and sentence structure checking. 

4) Could you draw some insights for practice?

We agree with the reviewer that this would be useful. We have now added a Conclusions section that aims to narrow down the results to two key take-home message points, one about the input types and one about the off-policy predictors of on-policy performance.

Reviewer 3 Report

The authors of this manuscript collected an image dataset of rural roads in Estonia that are used for the World Rally Championship, utilizing a variety of sensors. Their proposed solution incorporates an Ouster LiDAR, which combines both LiDAR and RGB camera imagery.

The paper presents new findings, including the validation of the effectiveness of LiDAR-images for complex driving scenarios. Overall, the manuscript is well-written and scientifically sound. Therefore, I highly recommend it for publication.

Author Response

We thank the reviewer for the positive feedback and remain available for further questions if they should arise. We have now further referenced sources in the introduction and more precisely specified the goals and conclusions.

Original review text:
The authors of this manuscript collected an image dataset of rural roads in Estonia that are used for the World Rally Championship, utilizing a variety of sensors. Their proposed solution incorporates an Ouster LiDAR, which combines both LiDAR and RGB camera imagery.

The paper presents new findings, including the validation of the effectiveness of LiDAR-images for complex driving scenarios. Overall, the manuscript is well-written and scientifically sound. Therefore, I highly recommend it for publication.

Reviewer 4 Report

The general topic of this paper is actual and of great interest: autonomous driving. The importance of autonomous driving is given by at least three aspects: - The complexity of car traffic is increasing and, soon, most drivers will be overwhelmed by the complexity of situations for which they will have to make decisions in split seconds. - Autonomous driving will substantially reduce traffic accidents caused by poorly trained, speeding, and careless drivers, etc. - By relieving the driver of the task of driving, that person will have time during car travel for other activities: leisure and even work. An essential aspect of autonomous driving is decision-making based on the data provided by the sensors, and this is the focus of the authors’ research. Both the neural network for decision-making and two types of optical sensors are considered in this research. The paper is the continuation of the authors' research in the field. The work is of the field research type, very important for this domain, naturally alongside lab research. The paper completes the conclusions of other authors in the field and is notable for its amplitude, respectively for the fact that the research took place in all environmental conditions of the year. A recommendation would be that at the beginning of chapter 2 "Methods" a short theoretical synthesis of the methods discussed below should be made. Another recommendation would be that in tables where a row is repeated identically (as in Table 1), it should be written only once. The References section is relevant, but to emphasize the thorough approach to the current state, it would be recommended to expand it with 5-8 more titles. I recommend placing the Reference section before the Appendices. Other observations: Row 116: “The rally tracks are curvy and and are not”. Please correct the doble word. Page 16, Table A5: “These values serve as the basis for correaltion”. Please correct the word “correlation”.

Author Response

We thank the reviewer for an overall positive review and remain available for further questions. We extracted the following actionable suggestions from the review.

A recommendation would be that at the beginning of chapter 2 "Methods" a short theoretical synthesis of the methods discussed below should be made. 

We thank the reviewer for suggesting this and agree that such an introductory paragraph makes the Methods section more easily readable. We have now improved our manuscript by adding this paragraph (lines 81-87).

Another recommendation would be that in tables where a row is repeated identically (as in Table 1), it should be written only once.

We have now experimented with repeated and removed entries. The authors do not mind it one way or the other, as the model architecture remains comprehensible. Therefore we have agreed to proceed with what the reviewer suggested and have modified Table 1 accordingly.

The References section is relevant, but to emphasize the thorough approach to the current state, it would be recommended to expand it with 5-8 more titles. 

We thank the reviewer for the attention put to this aspect. We have now additionally referenced some more recent works and expanded the referencing of some claims made in the introduction and methods sections.

I recommend placing the Reference section before the Appendices. 

We are open to changing the order of the sections if the editorial office agrees to this; the ordering was proposed by the template used.

Other observations: Row 116: “The rally tracks are curvy and and are not”. Please correct the doble word. Page 16, Table A5: “These values serve as the basis for correaltion”. Please correct the word “correlation”.

We thank the reviewer for pointing out these mistakes; we have now corrected these errors, as well as a number of other mistakes.

Round 2

Reviewer 1 Report

The authors have addressed all the required comments

Reviewer 2 Report

No further comments